# Wavevector multiplexed atomic quantum memory via spatially-resolved single-photon detection

Michał Parniak [1], Michał Dąbrowski[1], Mateusz Mazelanik[1], Adam Leszczyński [1], Michał Lipka[1] & Wojciech Wasilewski[1]

Parallelized quantum information processing requires tailored quantum memories to simultaneously handle multiple photons. The spatial degree of freedom is a promising candidate to facilitate such photonic multiplexing. Using a single-photon resolving camera, we demonstrate a wavevector multiplexed quantum memory based on a cold atomic ensemble. Observation of nonclassical correlations between Raman scattered photons is confirmed by an average value of the second-order correlation function $g^{(2)}_{\mathrm{S,AS}} = 72 \pm 5$ in 665 separated modes simultaneously. The proposed protocol utilizing the multimode memory along with the camera will facilitate generation of multi-photon states, which are a necessity in quantum-enhanced sensing technologies and as an input to photonic quantum circuits.

[1] Institute of Experimental Physics, Faculty of Physics, University of Warsaw, Pasteura 5, 02-093 Warsaw, Poland. Correspondence and requests for materials should be addressed to M.P. (email: michal.parniak@fuw.edu.pl) or to M.Dąb. (email: mdabrowski@fuw.edu.pl)

Multiplexing in optical fibers or in free-space is essential in modern techniques for high-throughput transmission[1,2]. Similarly, as quantum technologies mature, the necessity of multiplexing in photon-based quantum communication becomes clear[3] and much effort is devoted to various schemes exploiting spatial[4–12], temporal[13–18], or spectral[19,20] degrees of freedom. Utilization of many modes can finally allow efficient application of the Duan–Lukin–Cirac–Zoller (DLCZ) protocol[19,21–24] and offer nearly deterministic generation of multi-photon states[25–27] later applicable in quantum-enhanced sensing technologies[28,29] as well as optical quantum computation[30].

Regardless of substantial efforts, the task of achieving large number of modes remains a challenging endeavor especially in hybrid atom-photon systems. In purely photonic, memoryless systems, hundreds of modes have been obtained within the spatial domain of spontaneous parametric downconversion[31–33] or by means of frequency-time entanglement[18,34–38]. However, most applications such as the DLCZ protocol[21], enhanced photon generation[26,27,39], or even linear optical quantum computing[30] require or greatly benefit from a multimode quantum memory. For instance, the largest number of temporal modes used for photon storage in cavity-based quantum memory[35] is 30. Likewise, ensembles of dopants in crystals have been used to store externally generated photons[40] in up to 100 modes or otherwise generate them[23,24] in 12 modes.

Another mainstream trend is to build a multiplexed quantum repeater by splitting a trapped atomic ensemble into many cells[5,26]. The idea was recently realized in two dimensions achieving 225 modes[41]. These schemes however suffer from the limitation given by difficulty in trapping large ensembles as well as hinder heralded simultaneous excitation of all modes. In consequence, they are rendered useful only for the DLCZ quantum repeater[21], but neither for quantum imaging[42–44] nor enhancing rate of the photonic state generation[18,19,26,27].

The purpose of this paper is a demonstration of massive improvement in the number of modes processed by the quantum memory. The experimental realization is accomplished through multiplexing of angular emission modes of a single quantum memory[27] and by employing a spatially resolved single-photon detection. Our experimental setup generates photons in 665 pairwise-coupled modes, exploring the regime of multimode capacity with simultaneous extremely low noise-level achieved with stringent, spatially multimode yet simple and robust filtering. We use a single-photon resolving camera to measure both correlations and autocorrelation unambiguously proving quantum character of light. Note that throughout our results, we do not perform accidental or noise background subtraction—in contrast to any previous experiments with single-photon sensitive cameras[12,27,31,32]. We achieve the quantum memory lifetime of >50 μs, which combined with the multimode capacity invites real-time feedback processing of stored excitations[45] and paves the way toward promptly achieving fast generation of single- and multi-photon states[26,27].

## Results

**Multi-photon generation.** Here we propose the potential application of our scheme as a platform for multi-photon state generation. Figure 1 pictures a protocol utilizing the multi-pixel capability of the single-photon resolving camera to enhance generation of multi-photon states. The essential advantage over recently introduced quantum memory arrays[41] is simultaneous excitation and access to many modes. The protocol is being managed by a classical memory storing the wavevectors of registered photons and the which-mode information. This

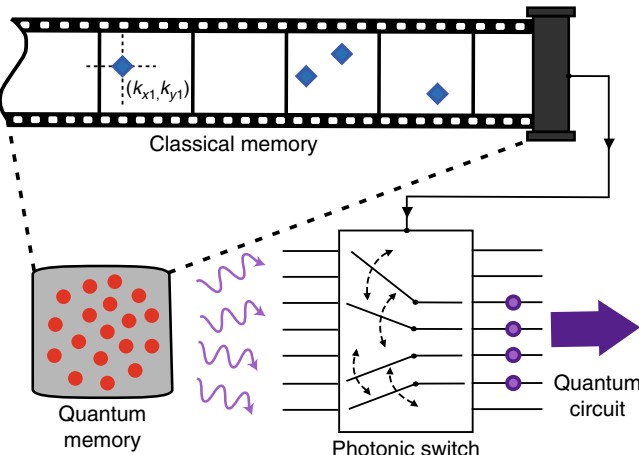

**Fig. 1** Proposal for single-photon spatial routing and multiplexing for multi-photon state generation. A single-photon resolving camera registers consecutive frames during the quantum memory write-in process. Each detection heralds creation of a spin-wave excitation in the atomic ensemble quantum memory with a wavevector determined by the position $(k_{xi}, k_{yi})$ at which $i$-th photon was registered. Acquisition and write-in continue until the desired number of excitations has been created. At this time a photonic switch is reconfigured to channel photons from conjugate directions stored in a classical memory and subsequently the readout pulse is applied to convert stored spin-wave excitations to the requested number of photons, which will be used later, e.g., in the quantum circuit

information is finally used to route the photons retrieved in the readout process from the quantum memory. A photonic switch is used to direct photons to a quantum circuit or to conditionally generate arbitrary states through multi-photon interference. Furthermore, since a small number of photons is generated per each frame, one can adapt in real-time the number of trials to create exactly the desired number of excitations in the quantum memory. By keeping the mean photon number per shot small, we virtually eliminate the malicious pairs in a single mode. This gives us an advantage over a simpler scheme[27] in which a single excitation shot is used. In that scheme, the mean number of photons may be controlled but the multimode thermal statistics severely limits the fidelity of generation of $n$-photon state. Extensions of our new proposal are numerous, including usage of spin-wave echos to conditionally manipulate the atomic excitations[46]. The experimental results presented below constitute the most essential step toward realization of the proposed protocol.

**Wavevector multiplexing.** For the quantum memory, we use an engineered atomic ensemble of cold rubidium-87 atoms at $T = 22 \pm 2$ μK generated within a magneto-optical trap (MOT) and cooled using polarization-gradient cooling (PGC) scheme, as depicted in Fig. 2. With the 1 cm-long cigar-shape ensemble of diameter $w = 0.6 \pm 0.1$ mm (taken as $1/e^2$ diameter of the atomic column density) containing $N = 10^8$ atoms, we achieve optical depth OD = 40, which limits the memory readout efficiency[17,47]. Quantum memory operates once atoms are released from MOT with the magnetic field gradients switched off. We prepare 70% of atoms in the $F = 1$, $m_F = 1$ state and the rest of atoms in the $F = 1$, $m_F = 0$ state through optical pumping. Atom-photon interface is achieved with two lasers: write, which is red-detuned from $5^2S_{1/2}$, $F = 1 \rightarrow 5^2P_{3/2}$, $F = 2$ transition and read laser tuned to $5^2S_{1/2}$, $F = 2 \rightarrow 5^2P_{1/2}$, $F = 2$ transition.

To generate the multimode, multi-photon state, we illuminate the ensemble with a write pulse containing $10^7$ photons

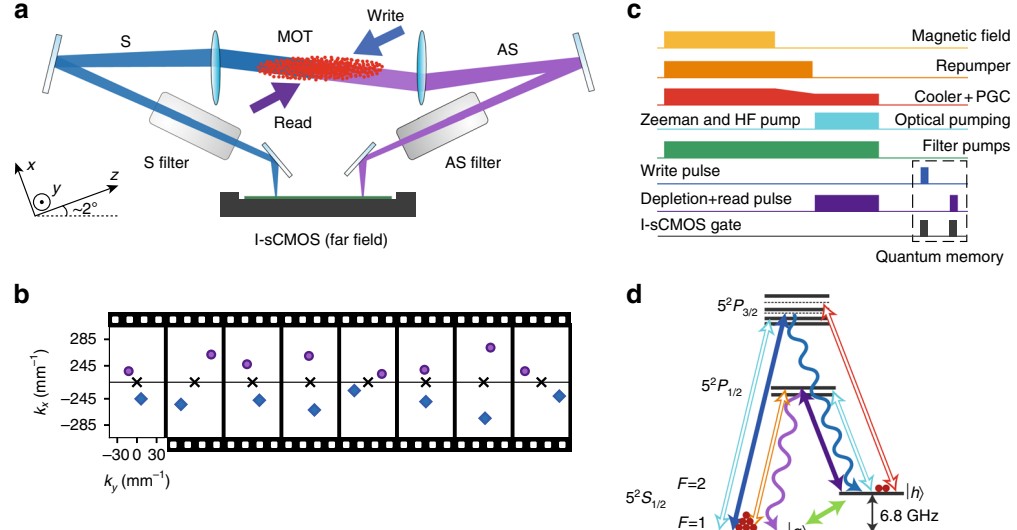

**Fig. 2** Experimental realization of the wavevector multiplexed quantum memory. **a** Schematic of the main part of the experimental setup. The atomic ensemble released from the magneto-optical trap (MOT) is illuminated with orthogonal circularly polarized write and read laser beams. Angles at which Stokes (S) and anti-Stokes (AS) photons (produced through Raman scattering) are emitted from the atomic ensemble are imaged on the single-photon resolving I-sCMOS (intensified scientific complementary metal-oxide semiconductor) sensor, composed of a sCMOS camera and an image intensifier. Optically pumped atomic cells (S and AS filters) filter out the residual laser light and stray fluorescence. **b** Example subsequent frames in the Stokes (bottom) and anti-Stokes (top) regions demonstrating correlated photon pairs in each camera frame. Note that while most frames will contain no photon or a photon only in a single region, almost all (>90%) frames with a coincidence event will contain a correlated photon pair for the detection probability of S photon $p_S = 1.2 \times 10^{-2}$. **c** Pulse sequence used in the experiment (see Methods for details) consists of trapping magnetic field switching, laser cooling, and optical pumping (with depletion) preparation stages, as well as short write-in and readout laser pulses (quantum memory stage) producing Stokes and anti-Stokes photons. **d** Atomic level configuration (colors correspond to the pulse sequence in **c**). In the process of write-in and readout, the spin wave (green arrow) is created and annihilated, respectively. The wavy arrows correspond to S (blue) and AS (purple) photons

with wavevector $\mathbf{k}_w$ tilted at an angle of 2° to the ensemble axis in the $xz$ plane. We take the axis defined by counter-propagating read and write beams as the $z$ axis of the frame of reference. Stokes (S) photons scattered in the Raman process are registered on the I-sCMOS camera located in the far-field with respect to the atomic ensemble. A scattered photon with a transverse wavevector $\mathbf{k}_S$ is accompanied by a collective atomic excitation (spin wave) with a spatial phase dependance: $N^{-1/2} \sum_{j=1}^{N} \exp\left(i\mathbf{K} \cdot \mathbf{r}_j\right)\left|g_1 \ldots h_j \ldots g_N\right\rangle$, where $\mathbf{K} = \mathbf{k}_S - \mathbf{k}_w$ is the spin-wave wavevector, $\mathbf{r}_j$ is the position of $j$-th atom, $\left|g_j\right\rangle$ $\left(\left|h_j\right\rangle\right)$ corresponds to the $5^2S_{1/2}$, $F=1$ ($F=2$) state, and the summation is carried over all $N$ atoms in the ensemble. To learn about the spin wave, we convert it to an anti-Stokes (AS) photon through resonant Raman scattering (readout process) with a read pulse with wavevector $\mathbf{k}_r$ containing $10^8$ photons. Wavevector of the AS photon is determined by the stored atomic excitation $\mathbf{k}_{AS} = \mathbf{K} + \mathbf{k}_r$. We estimate the readout efficiency as $\chi_R = 35 \pm 2\%$ (taken as the ratio of coincidence rate to S photons rate and accounted for losses). AS photons are registered on a separate region of the same I-sCMOS sensor.

Spatially insensitive filtering is essential for the memory to take advantage of its inherent multimode capability. Commonly used frequency filtering cavities[5,11,13] transmit only one spatial mode. To overcome this issue, we use two separate optically pumped hot rubidium vapor cells with buffer gas and paraffin coating. The cells are pumped by strong lasers during the cooling and trapping period of the MOT. Additional interference filters are used to separate stray pump laser light from single photons (see Methods for details).

Finally, photons originating from the atomic quantum memory are imaged onto the I-sCMOS sensor through a nearly diffraction-limited imaging setup. The sensor is located in the Fourier plane of the atomic ensemble. Positions of photons

registered on the camera are calibrated as transverse emission angles, directly proportional to transverse wavevector components. The I-sCMOS camera has the quantum efficiency of 20% and the combined average transmission of the imaging and filtering system is 40% (see Methods for details). The net efficiencies in S and AS arms are equal $\eta_S \approx \eta_{AS} \approx 8\%$.

**Data analysis**. The spatial degree of freedom provides an advantage over single-mode experiments[8,9]. If one considers each mode as a separate realization of the experiment, we are able to collect statistics at a rate of $3 \times 10^5$ effective experiments per s. This rate is very similar to what is obtained in single-mode experiments, however the multimode scheme offers much more versatility as increasing the memory storage time to many μs decreases the rate very insignificantly, contrasted with a dramatic drop of the rate in the single-mode experiments. For example, with 30 μs storage time, our effective experimental rate remains at 300 kHz, as it is anyway limited by the readout speed of the sCMOS camera. For the corresponding single-mode experiment, the absolute maximum stands at 33 kHz. With faster camera acquisition rate, the advantage of the multimode scenario would become overwhelming.

Here, to obtain proper statistics we have collected $10^7$ camera frames. For a pair of small conjugate square-shaped region of interests (ROIs) with side length $\kappa = 160 \ \text{mm}^{-1}$ and a net S photon detection probability of $4 \times 10^{-2}$, we register very few accidental coincidences, i.e., 90% of coincidences come from conjugate modes. This figure of merit changes with a mean photon number and thus the number of observed modes, as due to limited detection efficiencies, we will sometimes register a pair of photons from two different pairwise-coupled modes. For two conjugate ROIs with a side lengths $\kappa = 340 \ \text{mm}^{-1}$ (i.e., 43 mrad)

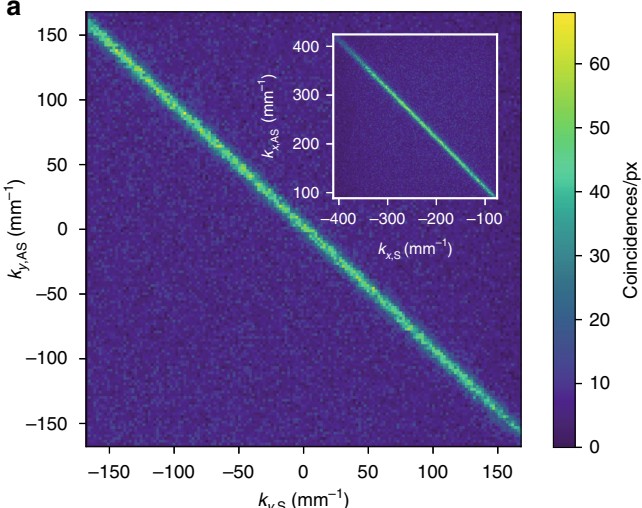

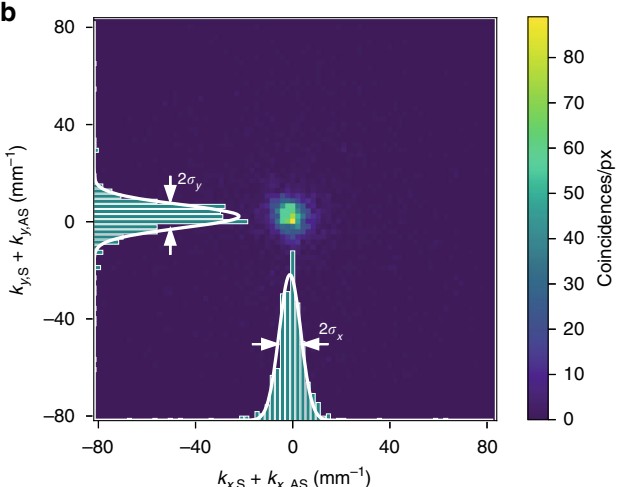

**Fig. 3** Spatial properties of the generated biphoton state. **a** All Stokes–anti-Stokes coincidences obtained from $10^7$ frames marked with their $k_y$ wavevector components ($k_x$ for the inset) for zero memory storage time, demonstrating high degree of momenta anti-correlation. Each plot disregards the perpendicular component. **b** Same coincidences counted in the center-of-mass variables ($k_{x,S} + k_{x,AS}$) and ($k_{y,S} + k_{y,AS}$). The central peak is fitted with a two-dimensional Gaussian to obtain its center and width. One-dimensional distributions correspond to cross-section counts selected for central pixels. Both in **a**, **b**, neither accidental nor noise background subtraction is performed

corresponding to a nearly full field of view composed of hundreds of modes, we have registered a total number of $1.6 \times 10^5$ coincidences of which $4.4 \times 10^4$ came from conjugate mode pairs.

Collection of photon counts with a multi-pixel detector requires new experimental and data analysis tools[48]. To verify the anti-correlation between momenta of S and AS photons both in $x$ and $y$ coordinates, we count the coincidences for each pair of camera pixels corresponding to wavevector coordinates ($k_{x,S}, k_{y,S}$) and ($k_{x,AS}, k_{y,AS}$). Figure 3a portrays the number of coincidences for a large field of view as a function of ($k_{y,S}, k_{y,AS}$) momenta summed over the $x$ coordinates. Notably, thanks to a very high signal-to-noise ratio, we do not subtract the accidental and noise background in contrast to hitherto schemes[12,27,31,32]. We observe a clear anti-correlated behavior that we model by the quantum amplitude of the generated Stokes–anti-Stokes photon pair in $y$

dimension, given by:

$$\Psi_y\big(k_{y,S}, k_{y,AS}\big) = \mathcal{N} \exp\left(-\frac{(k_{y,S} + k_{y,AS})^2}{2\sigma_y^2}\right), \quad (1)$$

where $\sigma_y$ is a correlation length in the $y$ dimension and $\mathcal{N}$ is a normalization constant. In turn, the number of coincidences is proportional to $\big|\Psi_y\big(k_{y,S}, k_{y,AS}\big)\big|^2$. An identical expression describes photons behavior in $x$ dimension—see inset in Fig. 3a. For the Gaussian-shaped atomic ensemble, the size of the emission mode should be related to ensemble transverse dimension $w = 0.6 \pm 0.1$ mm, corresponding to wavevector spread of $2/w = 3.3 \pm 0.5$ mm$^{-1}$ in the far field for light at the wavelength of S photons. To precisely determine the mode widths $\sigma_{x,y}$, in Fig. 3b, we plot the coincidences in terms of sum of wavevector variables. Gaussian fit yields values of $\sigma_x = 4.45 \pm 0.02$ mm$^{-1}$ and $\sigma_y = 4.76 \pm 0.02$ mm$^{-1}$ for $x$ and $y$ dimension, respectively. Consequently, we can consider the generated entangled state[10,12] to be nearly symmetrical in terms of $x$ and $y$ spatial dimensions. This wavevector spread is very close to the limit $2/w$ given by the diffraction at the atomic ensemble and confirms the quality of the imaging system for conjugate modes.

**Capacity estimation.** From the fundamental point of view, multimode states of light can be considered either as continuous-variable systems[49] or highly dimensional entangled states[50] offering large dimensionality of available Hilbert space and in turn providing high informational capacity. Estimation of the informational capacity of continuous-variable entangled states of light has attracted some attention of its own due to broad applications of such states[12,32]. Various measures of this capacity have been discussed, e.g., on the information-theoretical basis[51]. Here we estimate the number of independent mode pairs observed in S and AS arms using the Schmidt mode decomposition[52,53]. For a single-dimensional photon pair amplitude given by Eq. (1) and cropped to a finite region, we find a decomposition into the Schmidt modes as:

$$\Psi_y\big(k_{y,S}, k_{y,AS}\big) = \sum_{j=0}^{\infty} \lambda_j u_j\big(k_{y,S}\big) v_j^*\big(k_{y,AS}\big), \quad (2)$$

where $\lambda_j$ are singular values corresponding to contributions of each mode, while $u_j$ and $v_j^*$ are orthogonal sets of eigenfunctions. Effective number of independent mode pairs is given by $M = 1/\sum_{j=0}^{\infty} \lambda_j^4$, which may be also described in terms of modes given in an orbital-angular-momentum[7,8,50] or another orthogonal basis. We find the effective number of modes $M_{x,y}$ is well approximated by an inverse relation $M_{x,y} = 0.565 \, \kappa/2\sigma_{x,y}$ (see Methods for justification) and obtain $M_x = 26.7 \pm 0.1$ and $M_y = 24.9 \pm 0.1$. Finally, for the total number of modes $M$, which is the product of the number of modes in each spatial dimension, we get $M = M_x M_y = 665 \pm 4$.

**Nonclassical photon-number correlations.** The presented spatial correlations at a single-photon level require further analysis to confirm actual generation of multi-photon quantum states of light. Quantumness of the correlations (and hence the memory) may be assessed by looking at the second-order correlation function:[54]

$$g_{S,AS}^{(2)} = \frac{p_{S,AS}}{p_S p_{AS}}, \quad (3)$$

where $p_S$ and $p_{AS}$ are the probabilities of registering a Stokes and an anti-Stokes photon in their respective regions, while $p_{S,AS}$ is the Stokes–anti-Stokes coincidence probability. Since the

single-mode statistics of Stokes and anti-Stokes light are thermal[55], the maximum value of local $g_{S,S}^{(2)}$ and $g_{AS,AS}^{(2)}$ autocorrelation functions is 2. Consequently, a value of $g_{S,AS}^{(2)} > 2$ yields violation of the Cauchy-Schwarz inequality:[54]

$$R = \frac{\left[g_{S,AS}^{(2)}\right]^2}{g_{S,S}^{(2)} g_{AS,AS}^{(2)}} \leq 1 \qquad (4)$$

and thus proves nonclassical character of the generated state of light.

To perform the measurements, we utilize the photon-number resolving capability of the I-sCMOS detector[48]. We verify nonclassical photon-number correlations in many modes by selecting a set of ROIs in both S and AS arms and calculating $g_{S,AS}^{(2)}$ for all accessible combinations. Results presented in Fig. 4a, b clearly confirm the multimode capacity discussed in previous sections. For the experimental data presented in Fig. 4b, we obtain $g_{S,AS}^{(2)} = 72 \pm 5$ at the diagonal compared with $g_{S,AS}^{(2)} = 1.0 \pm 0.4$ for a set of uncorrelated regions, where the errors correspond to one standard deviation. Next, we select a single pair of square-shaped conjugate ROIs in S and AS arms. Figure 4c presents the measured $g_{S,AS}^{(2)}$ at $t = 0$ storage time for varying size of ROI with a constant photon flux per pixel. With the decreasing size of ROI, the S photon detection probability $p_S$ decreases and we observe $g_{S,AS}^{(2)}$ cross-correlation well above the classical limit of 2, which perfectly matches our theoretical predictions (see Methods for detailed theory of $g_{S,AS}^{(2)}$ measured with a multi-pixel detector). We compare this result with a maximum value achievable without noise in the AS arm as well as maximum theoretical value for two-mode squeezed vacuum state for the given $p_S$, achievable only if coherent spatial filtering (using e.g., single-mode fibers or cavities) is applied.

Even though we expect the photon statistics in S and AS arms to exhibit maximum values of autocorrelation functions of 2, to implicitly demonstrate violation of Cauchy–Schwartz inequality (4), we have performed additional measurements of $g_{S,S}^{(2)}$ and $g_{AS,AS}^{(2)}$ using a slightly modified experimental setup (see Methods for details). Due to inherently low number of S–S and AS–AS coincidences, we have increased the mean photon number in the S arm to 1.2 obtaining an average value of $R = 4.0 \pm 0.2$, significantly violating inequality (4) and proving both $g_{S,S}^{(2)}, g_{AS,AS}^{(2)} \leq 2$ (see Supplementary Fig. 2 for spatially resolved maps).

**Storage capabilities.** Cold atomic ensemble prepared in MOT typically offers µs up to ms coherence times, limited mainly by atomic motion, atom losses, and stray magnetic fields. We characterize the memory storage time by analyzing the $g_{S,AS}^{(2)}$ correlation function when the read laser is applied after a variable storage time $t$ following the write pulse. Figure 4d presents the average $g_{S,AS}^{(2)}$ calculated for 1000 pairs of correlated square-shaped ROIs with side length $\kappa = 21$ mm$^{-1}$ and $p_S = 1.9 \times 10^{-3}$ per entire ROI, each comprising approximately five modes. Data sets in Fig. 4d correspond to two different angles at which the photons were scattered, hence spin waves with different wavevectors—higher scattering angles (and thus spin waves with larger wavenumbers) correspond to shorter decay times. We observe a quantum-beating oscillation on a double exponential decay of correlations due to the presence of two types of spin waves arising as a result of imperfect optical pumping (see Methods for details). Due to the axial magnetic field of 36 mG, the two types of spin waves accumulate different phases over the storage time that leads to their constructive or destructive interference at the readout stage. We observe this interference effect as an oscillation with a Larmor period of $T = 2\pi/\omega = 19.5$ µs. For all spin waves, we

measure lifetimes larger than 50 µs. In particular for the clock-transition spin wave (between $F = 1$, $m_F = 1$ and $F = 2$, $m_F = -1$ states) with small $K_x = 100$ mm$^{-1}$, we obtain the lifetime of over 100 µs. The main source of decoherence is the random atomic motion governed by the Maxwell–Boltzmann velocity distribution[56]. The sharp drop in $g_{S,AS}^{(2)}$ in the very beginning (two initial experimental point) is attributed to increase of noise fluorescence as a result of an influx of unpumped thermal atoms into the interaction region. This noise might be eliminated by optical pumping of thermal atoms or by using a two-stage MOT with differential pumping. See Supplementary Fig. 3 for the measured temporal evolution of noise fluorescence.

## Discussion

We have demonstrated a quantum memory-enabled source of spatially structured nonclassical light based on a principle of wavevector multiplexing. Simultaneous operation on many collective atomic excitations allows us to generate a multimode quantum state of light. The memory preserves nonclassical correlations up to 50 µs and exhibits excellent noise properties, in contrast to the hitherto used warm-atomic vapor schemes[12,27]. Simultaneous detection using a state-of-the-art single-photon resolving camera is an ideal scheme to implement the enhanced photon generation protocols[26,27,45]. Additionally, a two-dimensional detector is both necessary and well-suited to the access high quantum information capacity of multimode states of light, which is unachievable with single-mode fibers[31]. Furthermore, simultaneous detection of the entire transverse field of view is essential in fundamental tests such as demonstration of the Einstein–Podolsky–Rosen paradox[57] without the Bell sampling loophole[32].

Our results clearly demonstrate the ability of multimode quantum memory to emit a single photon with high probability. In particular, we measured S photon detection probability of 0.21 and simultaneously extremely low probability of registering a photon per mode equal $3.8 \times 10^{-4}$ that drastically minimizes the probability of generating a photon pair in a single mode and proves memory efficacy in enhanced generation of photons. Excellent quality of single photons has been verified through measurements of $g_{S,AS}^{(2)}$ cross-correlation function. Our quantum memory also exhibits an excellent time-bandwidth product of >500, which is an important figure of merit in terms of probability of retrieving all the photons stored in the memory (Fig. 1), as well as prospective integration with time-bin multiplexing[26]. We envisage that hundreds of µs memory lifetime, contrasted with noise-free yet low storage-time solutions[58], and 100 ns operation time are excellent parameters when it comes to integration with fast electronic or photonic circuits for real-time feedback[25,59]. With these technical difficulties overcame, we expect that the proposed enhanced multi-photon generation protocol would be readily realizable. Integration of existing schemes[17] with readout efficiency $\chi_R$ of nearly 0.9 and the probability to generate $n$ photons equal $(\chi_R)^n$, will make our protocol highly competitive. Keeping a low probability of generating a photon–atomic excitation pair per mode $p \approx 0.01$, our setup can emit $n = pM \approx 6.6$ photons on average and thus could efficiently generate even six-photon states in the memory. Consequently, the number of modes $M$ places a fundamental yet possibly distant limit on generating multi-photon states.

The number of available modes is limited by the imaging system. In a cold atomic ensemble generated within a released MOT, we expect that the final limit for the number of modes will be set by the lifetime of long-wavevector spin waves as well as the phase-matching at the retrieval stage. To keep the lifetime within tens-of-microseconds regime the maximum scattering angles

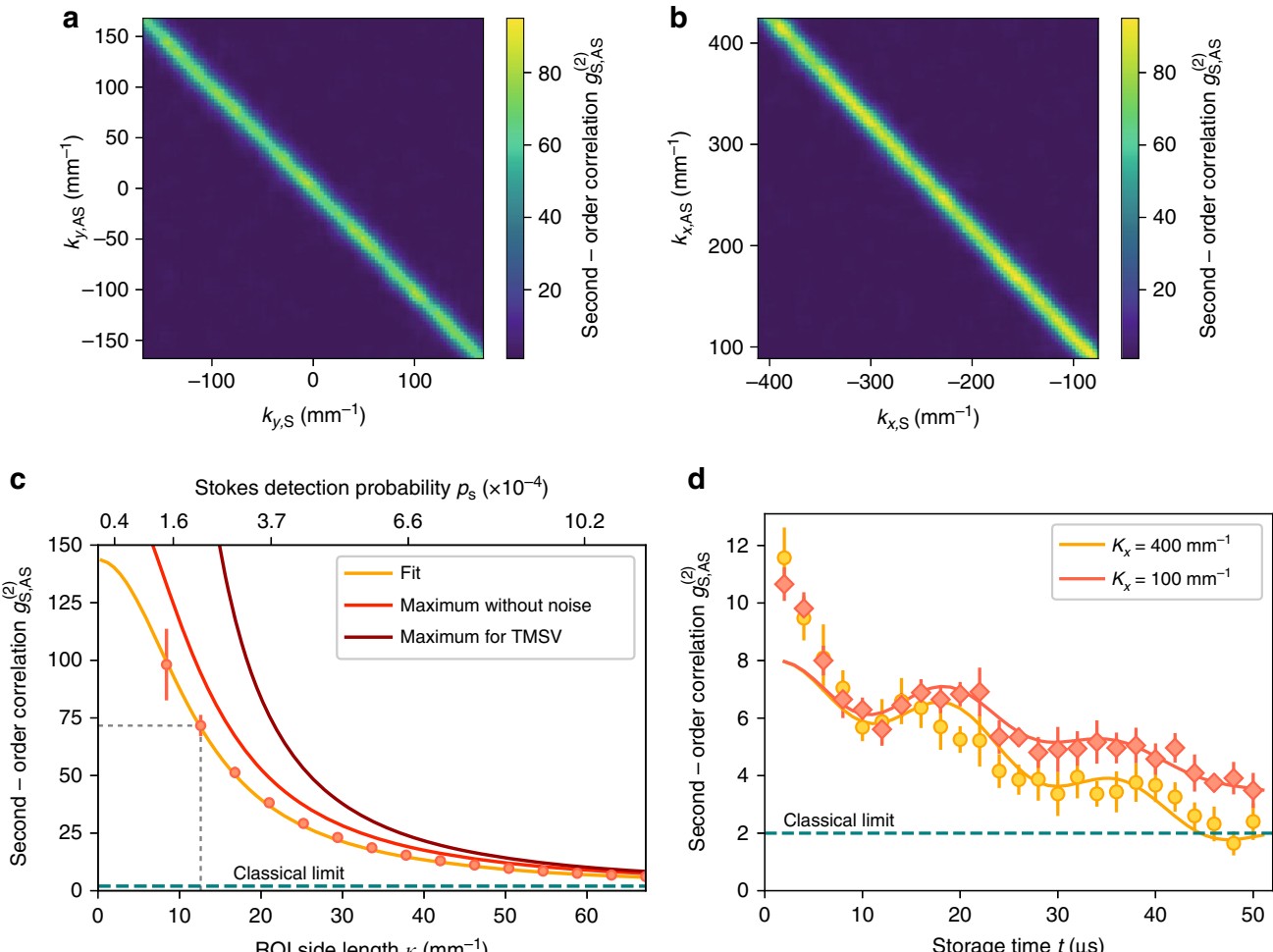

**Fig. 4** Nonclassical correlations of photons emitted from the quantum memory. **a**, **b** Second-order cross-correlation function $g_{S,AS}^{(2)}$ measured for different positions of ROI in S and AS arms, for zero memory storage time. Nonclassical correlations are observed only between conjugate modes, confirming the highly-multimode character of our quantum memory. Data corresponds to S photon probability $p_S = 2 \times 10^{-4}$ per ROI. Standard deviation error maps are included as Supplementary Fig. 1. **c** $g_{S,AS}^{(2)}$ for Stokes and anti-Stokes photons measured at $t = 0$ storage time using different sizes of ROI in the analysis. Smaller ROIs correspond to lower $p_S$ and consequently give higher values of $g_{S,AS}^{(2)}$. Our theoretical prediction for $g_{S,AS}^{(2)}$ calculated for the measured mode size closely adheres to experimental results (see Methods for details). Other curves correspond to the maximum value of $g_{S,AS}^{(2)}$ without noise in the AS arm and the maximum theoretical result for two-mode squeezed vacuum state (TMSV) with given probability $p_S$. Gray dashed lines mark the regime of operation used in the measurement shown in **a**, **b**. **d** Second-order correlation as a function of storage time, measured for two different angles of scattering corresponding to stored spin waves with different $K_x$. Data was taken with a higher than in (**a**–**c**) S photon detection probability of $p_S = 1.9 \times 10^{-3}$ and thus the value of the correlation function is smaller. Nonclassical correlations for spin waves with smaller wavenumber are confirmed for the storage time $t$ up to 50 μs. Theoretical model of the time evolution of $g_{S,AS}^{(2)}$ (solid lines, see Methods for derivation) exhibits good agreement with experimental data, except for the initial drop that we attribute to an increase of noise fluorescence of thermal atoms. Errorbars in **c**, **d** correspond to one standard deviation drawn from an ensemble of multiple conjugate region pairs

should be smaller than 6 degrees while the phase-matching happens to place a similar limitation[4,60]. We thus predict that the number of readily available modes may reach thousands under realistic experimental conditions. However, with novel spin-wave manipulation techniques[61] or by placing the atoms in an optical lattice[62] at least another order-of-magnitude improvement could be achieved, allowing our setup to serve as a universal platform for quantum state preparation.

While we have focused on the application of our quantum memory as a light source in multiplexed communication and computation protocols[26,27], our scheme is also perfectly matched to expedite quantum communication in free-space[1] or with multimode or multi-core fibers[2], quantum imaging, and image processing at the single-photon level, as well as quantum-enhanced metrology[28,29]. Spatial photon-number quantum correlations are readily applicable in quantum imaging techniques

and the memory capability could help quantum ghost imaging or sub-shot noise imaging along the way to practical applications[43,44]. Furthermore, the quantum-beat signal between two spin-wave excitations demonstrates the ability of our quantum memory to store a superposition of a few spin waves in many modes and paves the way toward manipulations within and between the Zeeman sublevels as well as with the spatial degree of freedom.

## Methods

**Experimental sequence**. Our experimental sequence depicted in Fig. 2c starts with trapping the atoms in MOT for 1.4 ms in an octagonal double-side AR coated glass chamber (Precision Glassblowing) with cooling laser red-detuned from $5^2S_{1/2}$, $F = 2 \rightarrow 5^2P_{3/2}$, $F = 3$ transition by 16 MHz followed by 300 μs phase of PGC (cooling laser detuning increased to 35 MHz by tuning the double-pass acousto-optic modulator) that brings the temperature from roughly 100 μK to $22 \pm 2$ μK. Fast MOSFET-based (metal-oxide semiconductor field effect transistor) coil switch

turns off the coil current in <5 µs from 125 A down to zero, and thus the MOT is turned off during memory operation. Small stray magnetic fields due to eddy currents take another 200 µs to decay but are compensated to nearly zero in the position of the atomic ensemble by a shorted compensation coils, which we verified by taking the free-induction decay measurements[63]. Compensation coils maintain a constant axial magnetic field of 36 mG during both the optical pumping and the memory operation stages. The cooling and trapping period is followed by 40 µs stage of optical pumping carried out by three lasers to ensure emptying of the memory state $|h\rangle$ and maximizing of Zeeman population in the $F = 1$, $m_F = 1$ state. Two lasers empty the $F = 2$ manifold: a strong pulse of the read laser (10 mW, depletion stage) as well as another laser (hyperfine pump—HF) tuned to $5^2S_{1/2}$, $F = 2 \to 5^2P_{1/2}$, $F' = 2$ transition, incident from four directions (with various polarizations) with a total power of 10 mW. Another laser (Zeeman pump) with a power of 7 mW is resonant to $5^2S_{1/2}$, $F = 1 \to 5^2P_{3/2}$, $F' = 1$ transition and transfers the population from $m_F = -1$ and $m_F = 0$ to $m_F = 1$ state of the $F = 1$ manifold. Dark period of 1 µs follows the optical pumping to ensure all light from lasers and from the atomic ensemble is extinguished. Next, a 100 ns write pulse is applied (left-circular polarization, red-detuned by 20 MHz from $5^2S_{1/2}$, $F = 1 \to 5^2P_{3/2}$, $F' = 2$ transition). Due to small detunings from respective energy levels, the influence of deleterious processes of readout (write-in) with the write (read) laser is negligible. After a variable memory storage time, a 200 ns read pulse (right-circular polarization, resonant with $5^2S_{1/2}$, $F = 2 \to 5^2P_{1/2}$, $F = 2$ transition) is applied. All lasers are locked to either cooler or repumper laser through a beat-note offset lock[64].

**Imaging**. The angles at which the photons are emitted from the atomic ensemble are imaged on the photocathode of the image intensifier with two separate (for S and AS) complex telescopes composed of six lenses, each having an effective focal length of $f_{eff} = 50$ mm. Total length of the single system is 2 m as optically pumped atomic filters need to fit along the photons path. The linear size of one pixel of the sCMOS camera corresponds to transverse wavevector size of 2.1 mm$^{-1}$ or angle of 265 µrad. Lens apertures limit our field of view to a ROI with $\kappa = 420$ mm$^{-1}$ or total solid angle of $\Omega = 2.76$ msr, which later determines maximum number of observable modes. The imaging system was calibrated with custom Ronchi rulings. The I-sCMOS device, composed of image intensifier (Hamamatsu V7090D) and an sCMOS sensor (Andor Zyla 5.5 MP) is gated (Photek GM300-3 gating module) only during write-in and readout of atomic excitations. The sequence is repeated at the rate of 500 Hz, which is limited by the frame rate of the sCMOS camera. The combined image intensifier gate duration is 400 ns for both write-in and readout stages. The probability of registering a dark count for the combined S and AS fields of view is ~$5 \times 10^{-3}$—much less than typical photon detection probability of 0.2–0.3.

**Filtering**. Two separate rubidium vapor cells are used to filter out stray write and read laser light from S and AS photons. The 10-cm-long cells are paraffin-coated and contain 99.4% isotopically pure $^{87}$Rb as well as buffer gases (Precision Glassblowing, 1 torr Kr for both S and AS filters) that keep the pumped atoms in the interaction region. The cells are pumped with 50 mW of resonant write laser light (with $5^2S_{1/2}$, $F = 2 \to 5^2P_{1/2}$ and $5^2S_{1/2}$, $F = 1 \to 5^2P_{3/2}$ for S and AS filters, respectively) in a double-pass configuration with collimated 1-cm-wide beams. The optical pumping is active at all times except when the image intensifier gate is open. Importantly, write and read lasers are filtered with Fabry-Pérot cavities before illuminating the ensemble to eliminate amplified spontaneous emission from laser diodes that would not be filtered with the hot atomic cell. Figure 5 presents a characterization of the S filter (we measured the comparable characteristics also for the AS filter). Both filters are characterized by OD >70 for the laser light and approx. sixty-five percent transmission for single photons generated inside the atomic quantum memory.

**Autocorrelation measurement**. Part of the experimental setup has been modified to allow measurement of autocorrelation functions $g^{(2)}_{S,S}$ and $g^{(2)}_{AS,AS}$. A high extinction-ratio Wollaston prism was placed in front of the image intensifier and a pair of half-wave plates was used to rotate the polarization of S and AS photons. The Wollaston prism split the photons into two beams (both for S and AS arm) at the 50:50 ratio in the vertical direction, so four distinct regions were observed on the camera (S1, S2, AS1, and AS2). After compensating for the change in angle of incidence due to refraction at the Wollaston prism, we have analyzed the correlations between regions S1–S2 and AS1–AS2 to obtain estimates of autocorrelation functions. The results are presented in Supplementary Fig. 2.

**Coincident counts**. Let us consider a collection of $M$ squeezed modes pairs. Assuming the probability $p$ of generating S photon in a single mode and efficiencies in the S and AS arms equal $\eta_S$ and $\eta_{AS}\chi_R$, respectively, with $\chi_R$ being the retrieval efficiency, we obtain the probability of registering a coincidence from any two conjugate modes $pM\eta_S\eta_{AS}\chi_R$. If we now consider a pair of square-shaped ROIs with the side length $\kappa$ containing $M$ modes for which we again assume the probability $p$ per mode to generate a photon pair, the coincidence rate is reduced, as not all coincidences will fall into the ROI. This effect is more pronounced for the smaller size of ROI. In particular, if we consider that the S photon is detected inside its respective ROI, we seek the probability that its conjugate AS photon will be

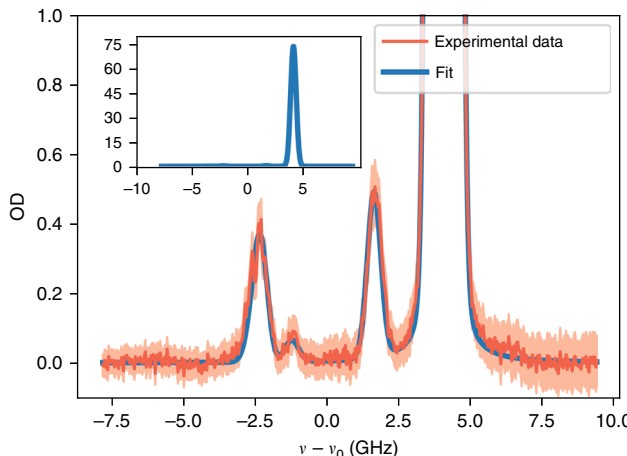

**Fig. 5** Characterization of the atomic filtering system for Stokes photons. Data points correspond to a measurement of absorption of the weak probe beam while fit is the theoretical prediction of OD based on a Voigt-profile absorption model including optical pumping. Inset shows the OD in a larger scale (only the fitted function), demonstrating very high attenuation of write laser light ($\nu - \nu_0 = 4.3$ GHz detuning). Simultaneously, high transmission of S photons ($\nu - \nu_0 = -2.5$ GHz detuning) is achieved. Similar characteristics were also obtained for the AS filter. Detuning is given with respect to the line centroid at $\nu_0$. Two more central absorption peaks corresponds to a residual amount of $^{85}$Rb in the filtering glass cell. Experimental errorbars correspond to one standard mean error derived from many collected spectra

detected in conjugate ROI (i.e., with conjugate center). We may calculate this probability by considering photon pairs distributed in momentum space according to Eq. (1). By considering S photons in the given ROI, we calculate the conditional probability $f(\kappa)$ of registering AS photon in the conjugate ROI in the AS arm, which gives us:

$$
\begin{aligned}
f(\kappa) &= \left( \int_{-\frac{\kappa}{2}}^{\frac{\kappa}{2}} dk_S \int_{-\frac{\kappa}{2}}^{\frac{\kappa}{2}} dk_{AS} \frac{1}{\sqrt{2\pi}\kappa\sigma} e^{-(k_S+k_{AS})^2/2\sigma^2} \right)^2 \\
&= \left( \mathrm{erf}\left(\frac{\sqrt{2}\kappa}{2\sigma}\right) + \sqrt{\frac{1}{2\pi}}\frac{2\sigma}{\kappa}\left(e^{-\kappa^2/2\sigma^2} - 1\right) \right)^2,
\end{aligned}
\tag{5}
$$

where squaring is due to the two-dimensional character of the problem. Finally, to estimate the net coincidence probability, we additionally consider the total number of accidental coincidences, which is very well approximated by a product of probabilities in S and AS arms $p_S p_{AS}$[12,24,56,65,66].

The net S–AS coincidence probability thus equals:

$$
p_{S,AS} = pMf(\kappa)\eta_S\eta_{AS}\chi_R + p_S p_{AS}.
\tag{6}
$$

**Second-order correlation**. We model the evolution of $g^{(2)}_{S,AS}$ correlation function following Zhao et al.[56], but including the effect of interference of different spin waves as well as the reduced number of coincidence counts due to incoherent spatial filtering. Finally, we end up with the following expression for the second-order correlation function:

$$
g^{(2)}_{S,AS} = 1 + \frac{p\eta_S\eta_{AS}\chi_R(t)f(\kappa)}{p\eta_S(p\eta_{AS}\chi_R(t) + \xi)},
\tag{7}
$$

where $\xi$ is a contribution of noise in the AS arm. The retrieval efficiency is modeled as an interference of two fields arising due to two atomic coherences by the following time-dependent expression:

$$
\chi_R(t) = \left| \alpha_1 \exp\left(-t^2/2\tau_1^2\right) + \alpha_2 \exp(i\omega t) \exp\left(-t^2/2\tau_2^2\right) \right|^2,
\tag{8}
$$

where $\alpha_1$ and $\alpha_2$ are contributions of spin waves between $F = 1$, $m_F = 1 \leftrightarrow F = 2$, $m_F = -1$ and $F = 1$, $m_F = 0 \leftrightarrow F = 2$, $m_F = -2$ transitions, respectively. The fit yields $\alpha_1 = 0.58$ and $\alpha_2 = 0.04$, clearly confirming dominant role of the clock-transition spin wave. The relative phase between the two spin waves changes as one of them accumulates additional phase due to a Zeeman energy shift $\hbar\omega = 2\pi\hbar \times 51$ kHz in the axial magnetic field of 36 mG. The lifetimes are bounded by wavevector-dependent decoherence rate $\Gamma_D = |\mathbf{K}|\nu$, with $\nu = \sqrt{\frac{k_B T}{m_{Rb}}} \approx 1.45$ cm s$^{-1}$.

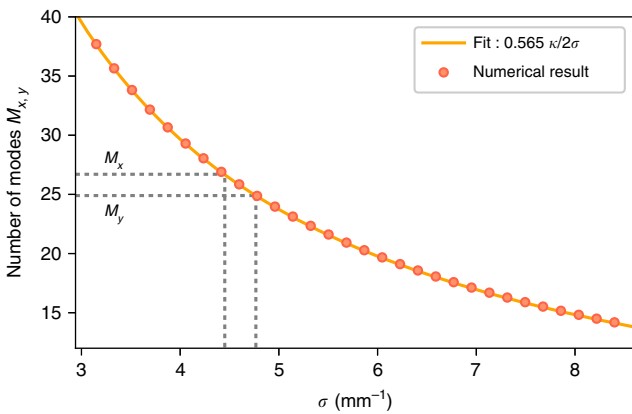

**Fig. 6** Result of the eigenmode decomposition for the number of modes. Dots represent the results from numerical decomposition, whereas the solid line is the simplified prediction of $0.565\kappa/2\sigma$. Dotted gray lines correspond to values of $\sigma_{x,y}$ we obtain in our experimental setup and corresponding numbers of modes $M_x$ and $M_y$

**Uncertainty estimation**. For the data presented in Fig. 4a to obtain a single $g^{(2)}_{S,AS}$ map, we have selected 100 regions in a column (oriented in the $y$-direction) in S and AS arms and calculated value of $g^{(2)}_{S,AS}$ for each pair of regions (note these regions are partially overlapping). We collect these results for 25 different conjugate positions of columns in the $x$-direction (i.e., $k_{x,S} + k_{x,AS} = 0$) and estimate mean and standard deviation. A completely analogous procedure was applied to obtain Fig. 4b and corresponding errorbars. Maps of standard deviation are included as Supplementary Fig. 1. To calculate the value of $g^{(2)}_{S,AS}$ for Fig. 4c, we have additionally averaged over all conjugate regions (corresponding to averaging over the diagonal in Fig. 4b) and inferring the errorbars (one standard deviation). Similar procedure was used to calculate values and standard deviations in Fig. 4d, however we used far less regions to average as we required that regions correspond to appropriate spin-wave wavevector $\mathbf{K}_x$. Furthermore, in this measurement less frames were collected for each point and thus we obtain relatively high uncertainty.

**Eigenmode decomposition**. To correctly determine the number of modes, we use a similar procedure as proposed by Law and Eberly[53]. Focusing on one dimension, we generate a normalized biphoton amplitudes according to Eq. (1), with various widths $\sigma$, on a square two-dimensional $k_{y,S} - k_{y,AS}$ grid. We numerically find the eigenmode decomposition of the generated matrix and calculate the number of modes according to Eq. (2). Figure 6 presents example of results for $\kappa = 420$ mm$^{-1}$ while the solid line corresponds to a fit of $A\kappa/2\sigma$ relation, which we verified numerically for various sets of parameters and obtained $A = 0.565$. Note that for a biphoton amplitude on a rectangular (non-square) grid numerical singular value decomposition might be used to give similar results.

**Data availability**. The data that support the findings of this study are available from M.P. upon reasonable request.

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

## Acknowledgements

We acknowledge insightful discussions about the MOT with L. Pruvost, G. Campbell, S. Du, G. Roati, M. Zawada, and W. Gawlik, assistance of R. Chrapkiewicz with optical system design, careful proofreading of the manuscript by K. T. Kaczmarek, M. Jachura, R. Chrapkiewicz, R. Łapkiewicz, and M. Semczuk, and generous support of K. Banaszek. The project has been funded by National Science Center (Poland) grant nos. 2015/19/N/ST2/01671, 2016/21/B/ST2/02559, and Polish MNiSW "Diamentowy Grant" Project nos. DI2013 011943, DI2016 014846.

## Author contributions

M.P., M.D., M.M. and A.L. performed the measurements and analyzed the data. M.P., M.D. and M.M. wrote the manuscript assisted by other authors. M.P., M.D., M.M., A.L., M.L. and W.W. contributed to building of the experimental setup. W.W. managed the project.

## Additional information

**Competing interests:** The authors declare no competing financial interests.

