## [Peer Review File · Nature Communications]

Reviewers' comments:

Reviewer #1 (Remarks to the Author):

The manuscript by Parniak et al. presents experimental observation of multi-mode quantum correlation between photons emitted from a quantum memory into different spatial modes. Stokes and anti-Stokes photons scattered with different k-vectors are detected and quantum correlations between them are verified. The same group in previous works including refs [24] and [42] has investigated the quantum nature of photons emitted in multi-spatial modes from mainly warm atomic systems. The current manuscript reports on observation of up to 665 correlated spatial modes of photons emitted from a cold atomic ensemble, a substantial enhancement important for multiplexing future quantum photonic networks. The results are certainly interesting and can have potential implications on future quantum communication technology and in principle worthy of publication in the Nature Communications Journal. However, I have questions and serious concerns regarding the way the data is presented and conclusion is drawn and would like authors to address the points raised below and provide necessary information so I can make an informative recommendation:

- Authors present result of cross correlation between Stokes and anti-Stokes photons in Fig4 and use it to verify the quantum nature of the emitted photons. It is important to know the values of auto-correlation functions $g_{\{as,as\}}$ and $g_{\{s,s\}}$ as well as the normalized cross correlation function $G = g_{\{s,as\}}^2 / g_{\{as,as\}} g_{\{s,s\}}$ including the error bar. This normalized correlation function is a figure of merit for characterizing quantum light sources violating Schwarz inequality. Also, properly accounting for error bar is important to put a bound on this inequality.

- Technical information: what is the size of the cigar-shaped cloud? Is the MOT turned off when measurement is performed? This information will help reader to evaluate the limitations of the system particularly because atomic motion is identified as the main source of decoherence.

- Technical information: what limits the optical pumping to a relatively poor efficiency of 70%? Is there any B-field applied during the pumping process? I am surprise that the optical pumping in such a cold atomic system is so far from the unity.

- In the first paragraph of page 3, authors compare the single mode and multimode regimes of operations and claim that extending the storage time does not significantly affect the collection rate.

This requires further explanation and some theoretical justification. Shouldn't the decoherences during the storage time affect the single mode and multimode the same way? Are the dominant sources of decoherence different in each case?

- In Fig 4d, the sharp increase in $g(2)$ is attributed to the unpumped atoms entering the interaction region. If this is the reason, why this was not simply avoided by using a pump with larger area to cover the entire cloud? Does the rate of decrease in $g(2)$ at early times agrees with the demultiping rate due to the "readout" pulse? Is this observation confirmed by enhanced $g_{\{s,s\}}$ and $g_{\{as,as\}}$ at short storage times?

- What is the storage time at which the data in Fig4c is plotted? If data is measured at zero storage time, Fig4d suggests the $g(2)$ can be contaminated by unpumped atoms. Is this the case?

- How are the error-bars calculated for different plots?

- In a paper by Ding published in Nat. Comm. in 2013 [DOI: 10.1038/ncomms3527], quantum storage of single photons with high-order spatial mode has been shown that should be cited.

Reviewer #2 (Remarks to the Author):

Parniak et al report realization of a multiplexed version of the Duan-lukin-Cirac-Zoller scheme. There have been a number of previous related demonstrations. Yet the present work is impressive

as it surpasses the prior attempts in the degree to which the goals of mutiplexing are being advanced. Namely they demonstrate a large - 665 - number of memories involved together with an approach that is conducive to further increase in this number while retaining high-quality quantum correlations. Their method is based on using distinct angular emission modes that are being imaged onto a single-photon-sensitive camera. This is an approach that should be of interest to a broad community in quantum optics. Other aspects of the experiment, such as multi-mode filtering using atomic cells will also be of use to other researchers.

I recommend publication in Nature Communications.

Reviewer #3 (Remarks to the Author):

Review of “Wavevector multiplexed quantum memory as a universal platform for quantum state preparation” by M. Parniak et al.

Parniak et al. present a scheme for using the spatial degree of freedom to achieve multiplexing in a DLCZ-type quantum memory. They measure spatial correlations between pairs of photons generated by Stokes and anti-Stokes scattering from ultracold rubidium atoms and are able to resolve an impressive 665 distinct spatial modes. Importantly, the second photon is “stored” in the quantum memory for several microseconds which gives time for feed-forward techniques in potential multiplexing applications. The paper is clearly written and the results are persuasive.

The major concern with the paper is that it promises wine but serves water. The implementation of Figure 1 would be a major step worthy of publication. However, the actual experiment is not what is shown in that figure. The paper details what is effectively an experiment that has been done before many times (in many different systems) but with a single-mode fibre coupler replaced by a camera. The presented experiment will be of interest to specialists in the quantum memory field though it may not be to the wider quantum photonics community. I suggest the authors endeavour to address this concern perhaps by reframing the paper or by explaining how the progress presented is larger than this reviewer understands.

There are a number of minor issues that should also be addressed before publication:

- 1) The primary application of this technique is to increase the photon generation rate while maintaining a low number of photons per mode. This, in turn, reduces the probability of accidentally generating multiple photons in any given mode which is nicely demonstrated in this work. However, another potential application (not demonstrated here) discussed in figure 1, and elsewhere in the document, is to use active feed-forward to generate multi-photon states. This is an extremely interesting idea, but I was wondering if the authors could comment on the effect the limited read efficiency (~30%) would have on such a technique?

2) Following on from this, on line 357-9 the authors refer to the time-bandwidth product and its relevance for time-bin multiplexing. I do not completely understand what is proposed here. Does this refer to the scheme outline in figure 1 where multiple write pulses used to generate multiple Stokes photons before a single read pulse retrieves all of the anti-Stokes photons? If so, perhaps a reference to figure 1 would help remind the reader.

3) A final point on the multi-photon generation scheme. If this is to be successful, then it is very important that the write pulse does not undergo anti-Stokes scattering. Otherwise the excitation generated by one write pulse will be “read out” by subsequent write pulses rather than by the read pulse. I imagine that anti-Stokes scattering of the write is prohibited by selection rules by choosing appropriate circularly-polarized light? If so, this should be clarified. And what effect does the imperfect optical pumping have on this?

4) In the Introduction, the authors provide a good background on the various methods for multiplexing single photon sources and QKD protocols. However, they have overlooked some important work from the Calgary group and would suggest the following two papers be referenced: Sinclair et al. DOI:10.1103/PhysRevLett.113.053603 and Grimau Puigibert et al. arxiv.org/abs/1703.02068.

5) Line 135: The states h and g are not defined until later in the document leading to some potential confusion

6) Line 257: similarly the parameter κ is not defined until the “storage capabilities” section

7) Figure 3a,b. Z-axis needs units. Coincidences per second?

Reviewers' comments and author's responses:

Reviewer #1:

We thank the Reviewer for finding our results interesting and deeming our manuscript publishable in Nature Communications as well as for pointing out several problems and suggesting additional research. We believe that our corrections inspired by these comments have led to substantial improvement of the manuscript and its scientific merits. Please find below detailed answers to all the comments along with descriptions of new measurements made.

The manuscript by Parniak et al. presents experimental observation of multimode quantum correlation between photons emitted from a quantum memory into different spatial modes. Stokes and anti-Stokes photons scattered with different k-vectors are detected and quantum correlations between them are verified. The same group in previous works including refs [24] and [42] has investigated the quantum nature of photons emitted in multi-spatial modes from mainly warm atomic systems. The current manuscript reports on observation of up to 665 correlated spatial modes of photons emitted from a cold atomic ensemble, a substantial enhancement important for multiplexing future quantum photonic networks. The results are certainly interesting and can have potential implications on future quantum communication technology and in principle worthy of publication in the Nature Communications Journal. However, I have questions and serious concerns regarding the way the data is presented and conclusion is drawn and would like authors to address the points raised below and provide necessary information so I can make an informative recommendation:

Question 1.1

Authors present result of cross correlation between Stokes and anti-Stokes photons in Fig. 4 and use it to verify the quantum nature of the emitted photons. It is important to know the values of autocorrelation functions $g^{(2)}_{\{AS,AS\}}$ and $g^{(2)}_{\{S,S\}}$ as well as the normalized cross correlation function $G = (g^{(2)}_{S,AS})^2 / (g^{(2)}_{AS,AS} g^{(2)}_{S,S})$ including the error bar. This normalized correlation function is a figure of merit for characterizing quantum light sources violating Schwarz inequality. Also, properly accounting for error bar is important to put a bound on this inequality.

Answer 1.1

We agree that the violation of the Cauchy-Schwarz inequality $R > 1$ where $R = (g^{(2)}_{S,AS})^2 / (g^{(2)}_{AS,AS} g^{(2)}_{S,S})$ is essential to demonstrate quantum photon-number correlations. Quite often it is assumed that the values of autocorrelation functions $g^{(2)}_{S,AS}$ and $g^{(2)}_{S,S}$ can reach a maximum value of 2, as the statistics in both fields are thermal and there is no particular reason to expect super-bunching. Furthermore, since our experiment operates at a very low light level, there is no possibility of detector saturation that would result in an overestimation of $g^{(2)}_{S,AS}$. From this argumentation stems the simplified quantum-classical limit of $g^{(2)}_{S,AS} = 2$, where $g^{(2)}_{S,AS} > 2$ indicates quantum photon-number correlations regime.

In our original experiment the values of autocorrelation functions were not measured, as the camera we use does not allow registering many photons in one spot - effectively a small area around one detected photon may be considered inactive. Consequently, we were not able properly determine values of $g^{(2)}_{AS,AS}$ and $g^{(2)}_{S,S}$.

Nevertheless, inspired by this insightful comment, we decided to modify the setup to allow us to measure autocorrelations. A Wollaston prism was placed in front of the image intensifier and polarization of light was adjusted to split each S and AS photons into two separate regions (S1, S2, AS1, AS2) on the camera with a 50:50 ratio. By registering coincidences between S1 and S2 as well as AS1 and AS2 we are able to determine values of autocorrelation functions. We believe that these results are a first measurement of this kind, i.e. using a camera to look both at correlations and autocorrelations.

We have performed this measurement in a slightly different regime of 1.2 Stokes photons registered on average per the entire S detection area, as simultaneous detection of 4 regions significantly limits the camera frame-rate and the number of S1-S2 and AS1-AS2 coincidences strongly depends on the mean photon number.

Even in this regime we have been able to significantly violate the Cauchy-Schwartz inequality. Plots of correlations function and the R parameter are attached below. At the diagonal we found an average of $R=4.0\pm 0.2$. For uncorrelated regions we inferred $R=0.68\pm 0.06$. We also include the plots as Supplementary Figure 2. Please see Answer 1.7 below as well as an additional section *Autocorrelation measurement in Methods* (see also Supplementary Figure 1) for detailed description of the error analysis methods we used. We envisage that in the low-photon regime the violation of the CS inequality would be much higher.

Question 1.2

Technical information: what is the size of the cigar-shaped cloud? Is the MOT turned off when measurement is performed? This information will help reader to evaluate the limitations of the system particularly because atomic motion is identified as the main source of decoherence.

Answer 1.2

The size of the atomic cloud is described in the original version of the manuscript, in line 112: *With the 1 cm-long cigar-shape ensemble (..)* as well as in line 218: *(..) ensemble transverse dimension $w = 0.6 \text{ mm}$ (..)*. In the revised version of the manuscript we put both parameters in one sentence, (line 124 revised manuscript numbering): *With the 1 cm-long cigar-shape ensemble of diameter $w = 0.6 \text{ mm}$ (taken as $1/e^2$ diameter of the atomic column density) (..)*.

As is depicted in Figure 2c, the MOT (magnetic field and laser beams) are turned off when the quantum memory operation starts. Please refer to the line 128: *Quantum memory operates once atoms are released from MOT with the magnetic field gradients switched off.* In particular, in the *Experimental sequence* section of *Methods* we added the direct information in line 447: *(..) and thus the MOT is turned off during memory operation.*

Question 1.3

Technical information: what limits the optical pumping to a relatively poor efficiency of 70%? Is there any B-field applied during the pumping process? I am surprised that the optical pumping in such a cold atomic system is so far from the unity.

Answer 1.3

The efficiency of optical pumping to $F=1, m_F=1$ of 70% was intentionally selected by using a short pump pulse to allow some of the atoms to reside in the $F=1, m_F=0$ state and later observe interference of two spin-waves (depicted in Figure 4d). At the same time we have used strong lasers to optically pump the atoms out of the $F=2$ manifold thus eliminating any spurious excitations that could result in additional noise present in the anti-Stokes signal.

During both the optical pumping stage and the memory operation a constant axial magnetic field of 36 mG is applied to set the quantization axis. We have added this information to line 453: *Compensation coils maintain a constant axial magnetic field of 36 mG during both the optical pumping and the memory operation stages.*

Question 1.4

In the first paragraph of page 3, authors compare the single mode and multimode regimes of operations and claim that extending the storage time does not significantly affect the collection rate. This requires further explanation and some theoretical justification. Shouldn't the decoherences during the storage time affect the single mode and multimode the same way? Are the dominant sources of decoherence different in each case?

Answer 1.4

The difference in the collection rate is the effect of inherent sequence type that differs in the single-mode and multimode case. In the single-mode case (or any experiment using fiber-coupled avalanche photodiodes) the MOT cloud is prepared and subsequently the write-read sequence is repeated at most ~ 3000 times [Nat. Commun. **7**, 13514 (2016)]. The number of repetitions is limited by expansion and free-fall of the cloud. Typically a single write-read sequence is less than 1 μ s long. If a multi-microsecond storage time is introduced, this severely limits the number of repetitions and thus the experimental rate.

In the spatially multimode case only one shot of the experiment is performed per one MOT cloud, since the camera currently offers frame-rate of a few hundred Hz. Consequently, storage times up to 100 μ s do not alter the experimental sequence rate significantly, and we can still benefit from multiple spatial modes, which constitutes an advantage over the single-mode case.

The explanation in the manuscript now reads (line 193): *(..) our effective experimental rate remains at 300 kHz, as it is any way limited by the readout speed of the sCMOS camera.* We also reframe further discussion to elucidate better understanding (see line 195): *For the corresponding single-mode experiment the absolute maximum stands at 33 kHz. With faster camera acquisition rate the advantage of the multimode scenario would become overwhelming.*

While the above discussion neglects the effects of decoherence, we expect the single-mode and multimode case to be influenced by decoherence in the same way, as long as the spin-wave wavevectors used are short enough to offer tens-of-microseconds lifetimes.

Question 1.5

In Fig. 4d, the sharp increase in $g^{(2)}$ is attributed to the unpumped atoms entering the interaction region. If this is the reason, why this was not simply avoided by using a pump with larger area to cover the entire cloud? Does the rate of decrease in $g^{(2)}$ at early times agrees with the depumping rate due to the “readout” pulse? Is this observation confirmed by enhanced $g^{(2)}_{S,S}$ and $g^{(2)}_{AS,AS}$ at short storage times?

Answer 1.5

In the experiment we use a relatively wide pump beams to cover the entire cold atomic cloud. Consequently the F=2 manifold is emptied with high efficiency. However, the vacuum chamber also contains a number of room-temperature untrapped atoms. The warm atoms that are in the volume covered by pump beams get optically pumped as well and reside in the F=1 manifold. However, after several microseconds unpumped warm atoms from outside the volume covered by the pump beams enter the interaction region. Since some of them reside in the F=2 manifold, they contribute to unwanted noise photons in the readout. We have verified this by taking a measurement of read photons without the write-in process, which we believe is (in this particular case) simpler and more direct evidence than autocorrelation measurement (see also Answer 1.1). Very similar results are obtained with or without the cooling and trapping mechanisms enabled.

The above figure presents the number of noise photons in the readout per one mode (blue dots). This is compared with the same figure of merit without the read beam (green dots). At zero storage times the noise is much smaller, approaching the dark counts level, than after several microsecond storage. Time-scale of this effect agrees well with the RMS thermal velocity of ~ 1100 m/s and pump beam diameter of 5 mm. Note that the slight oscillation is due to partial spin polarization of room-temperature atoms. We would like to stress that this is rather a technical limitations and could be solved by lowering the pressure of rubidium gas (which would require a two-stage MOT to keep the optical depth high) or by additional optical pumping of room-temperature atoms.

We have included the above figure as Supplementary Figure 3 and refer to it in line 366: See Supplementary Figure 3 for the measured temporal evolution of noise fluorescence.

Question 1.6

What is the storage time at which the data in Fig 4c is plotted? If data is measured at zero storage time, Fig 4d suggests the $g^{(2)}_{S,AS}$ can be contaminated by unpumped atoms. Is this the case?

Answer 1.6

The storage time for which the data in Figure 4c are plotted is $t=0$, which corresponds to the readout process starting immediately after the write-in. We add this information in line 305: *Figure 4c presents the measured $g^{(2)}_{S,AS}$ at $t=0$ storage time for varying size of ROI with a constant photon flux per pixel, as well as in the Figure 4c caption: $g^{(2)}_{S,AS}$ for Stokes and anti-Stokes photons measured at $t=0$ storage time using different sizes of ROI in the analysis.* For $t=0$ storage time the $g^{(2)}_{S,AS}$ measurement is very little contaminated by fluorescence from unpumped atoms, as is described in the Answer 1.5 above.

Question 1.7

How are the error-bars calculated for different plots?

Answer 1.7

In general, since the camera allows collection of many correlated regions at once, we have treated many pairs of regions as a statistical ensemble and inferred mean values and variances of correlation functions.

For Figure 4a to obtain a single $g^{(2)}_{S,AS}$ map we have selected 100 square regions in a column (oriented in y direction) in S and AS arms and calculated value for $g^{(2)}_{S,AS}$ for each pair of regions. Note that these regions are partially overlapping. To obtain statistics, we average over 25 different conjugate positions of columns in x-direction (so that $k_{x,S}+k_{x,AS}=0$). A completely analogous procedure was applied to obtain Figure 4b and corresponding Supplementary Fig. 1b. Please find below corresponding maps of standard deviation obtained in each point for Figures 4a and 4b (now attached as Supplementary Figure 1). From these we find typically a relative statistical errors of 30% at the diagonal and over 100% for uncorrelated regions due to very low number of uncorrelated counts. By averaging the diagonal of the plot we find $g^{(2)}_{S,AS}=72\pm 5$ and for a collection of 400 uncorrelated regions we find $g^{(2)}_{S,AS}=1.0\pm 0.4$, where the errorbars here correspond to one standard deviation. We have corrected the value given in the abstract to $g^{(2)}_{S,AS}=72\pm 5$ as well as added this information in line 301 of the revised manuscript. Note that as in Fig. 4c even higher values of $g^{(2)}_{S,AS}$ are achievable with smaller region-of-interest size, however than the uncertainty becomes high.

To calculate the value of $g^{(2)}_{S,AS}$ for Figure 4c, we have additionally averaged over all conjugate regions corresponding to averaging over a diagonal in Figure 4b and inferring the variance. Errorbars correspond to one standard deviation.

Similar procedure was used to calculate values and errorbars in Figure 4d, however we used far less regions to take the average as we required that the regions correspond to appropriate spin-wave wavevector \mathbf{K}_x . Furthermore, in this measurement less frames were collected for each point and thus we obtained higher uncertainty.

As plots in Figures 3a and 3b present merely coincidences registered in the experiment, no uncertainty is given, however we may approximately assume Poissonian statistics for each pixel if the mean coincidence probability is to be estimated.

For Fig. 5 multiple spectra were collected and the errorbars correspond to one standard mean error. This information has been added to figure caption.

A new section *Uncertainty estimation* in *Methods* was added that contains essential information on error analysis.

Question 1.8

In a paper by Ding published in Nat. Comm. in 2013 [Nat. Commun. 4, 252 (2013)], quantum storage of single photons with high-order spatial mode has been shown that should be cited.

Answer 1.8

The paper by *Ding et al.* was cited in the previous version of the manuscript as reference [45] but unfortunately we made a mistake in citation export. We apologize for that and cite the paper by *Ding et al.* in the correct form as reference [7] in the revised version of the manuscript.

Reviewer #2:

We appreciate the Reviewer positive reception of our manuscript and thank for the recommendation. We believe that the revised version should spark even greater interest in the community.

Parniak et al report realization of a multiplexed version of the Duan-Lukin-Cirac-Zoller scheme. There have been a number of previous related demonstrations. Yet the present work is impressive as it surpasses the prior attempts in the degree to which the goals of multiplexing are being advanced. Namely they demonstrate a large - 665 - number of memories involved together with an approach that is conducive to further increase in this number while retaining high-quality quantum correlations. Their method is based on using distinct angular emission modes that are being imaged onto a single-photon-sensitive camera. This is an approach that should be of interest to a broad community in quantum optics. Other aspects of the experiment, such as multi-mode filtering using atomic cells will also be of use to other researchers. I recommend publication in Nature Communications.

Reviewer #3:

We thank the Reviewer for the positive remarks as well as for the constructive criticism of our manuscript. As we believe, the main concern with our results was the interest to a broad community as well as the question of a true advantage of using a camera over optical fibers. Below we have included a detailed discussion of these issues with references to amended sections of the manuscript and response to all of the minor comments. We believe that our answer will convince the Reviewer of greater significance of these results and the substantial changes made to various parts of the manuscript will prove to be helpful for any future readers.

Parniak et al. present a scheme for using the spatial degree of freedom to achieve multiplexing in a DLCZ-type quantum memory. They measure spatial correlations between pairs of photons generated by Stokes and anti-Stokes scattering from ultracold rubidium atoms and are able to resolve an impressive 665 distinct spatial modes. Importantly, the second photon is “stored” in the quantum memory for several microseconds which gives time for feed-forward techniques in potential multiplexing applications. The paper is clearly written and the results are persuasive.

Question 3.1

The major concern with the paper is that it promises wine but serves water. The implementation of Figure 1 would be a major step worthy of publication. However, the actual experiment is not what is shown in that figure. The paper details what is effectively an experiment that has been done before many times (in many different systems) but with a single-mode fibre coupler replaced by a camera. The presented experiment will be of interest to specialists in the quantum memory field though it may not be to the wider quantum photonics community. I suggest the authors endeavour to address this concern perhaps by reframing the paper or by explaining how the progress presented is larger than this reviewer understands.

Answer 3.1

Even though generation of multi-photon states is indeed not realized in our work yet, we believe that the idea proposed in the Introduction and in Fig. 1 sets our scheme in an important context and acts as a beacon for future research, also for other groups. The reason why significant space is devoted to this topic is the fact that it constitutes a new idea, while the idea of enhanced single-photon generation has been presented before and we believe that the given references suffice (see Refs. [18,19,26,27,35]). We hope that the Introduction after amendments will not be confusing and clearly distinguishes our experimental results from the proposed protocol. Below we argue that the presented results are in fact first of their kind and discuss the profound differences between using a single-photon resolving camera and single-mode fibers. We believe that using a camera is a complementary approach to single-mode fibers and opens avenues towards new experiments.

With that said, our experiment realizes the most important and previously undescribed part of the proposed protocol. The experimental implementation of faster single-photon generation clearly demonstrates the feasibility and the potential of the combination of a multimode quantum memory with a single-photon resolving camera.

Feasibility of the proposed scheme

The remaining obstacle which is the realization of a real-time feedback is a task which we believe could be accomplished in the near future. The main technical problem of the implementation of a real-time feedback loop in the camera system is the acquisition time of the present solution (about 2 ms) and the processing time of another 10 ms which in combination are above the presented memory lifetime. A straightforward solution would be to use a memory with longer lifetime. As the optical-lattice memories with dynamic decoupling reach multi-second lifetimes [Phys. Rev. A **87**, 031801(R) (2013)], current speed of processing would be enough to allow real-time feedback. On the other hand, we expect that the integration of faster electronics with the sCMOS sensor of our intensified camera will allow processing significantly shorter than even the current memory lifetime.

Novelty and potential applications of presented results

As far as the novelty of the presented experimental results is concerned, our manuscript shows the first demonstration of the generation part (Stokes and correlated anti-Stokes scattering) of the DLCZ protocol using a camera sensitive to single photons. While perhaps similar results could be achieved with a large array of single-mode fibers and avalanche photodiodes, such experiment has not been described in the

literature, as other related works utilized at most only two modes at a time (see [Nat. Commun. **8**, 15359 (2017)] or [Opt. Express **17**, 13639 (2009)]).

Our experimental setup is designed specifically to be compatible with the camera measurements, including matched systems for imaging and filtering. New methods of data analysis are implemented to process the spatially-resolved photon count data. Calibration of our imaging setup with classical light allows us to measure the size and shape of the transverse spatial mode of photons emitted from the quantum memory. Single mode fibers as well as frequently used filtering cavities hinder such possibilities. This is crucial in the perspective of scalable solutions of the DLCZ protocol for quantum communication purposes, as well as allows us to demonstrate the interference of different spin waves stored in the same volume of atoms.

Furthermore, camera measurements are insensitive to the precise position of the single fibers or array of fibers. This is a crucial point in the situation in which mode-matching starts to play a key role in the experiment. Mismatch of the photonic mode and the detection mode of the optical fiber will effectively lower the detection efficiency significantly. Usage of a camera avoids this problem and provides an exquisite experimental flexibility. In particular, the mode shape and size may be precisely characterized (as demonstrated in our manuscript) and various configurations of imaging (e.g. far-field or near-field) may be easily selected.

Fundamental advantages and prospective related experiments

The fundamental informational advantage of using camera over scanning detectors is a separate issue. A single-photon resolving camera allows the observation of full field of view simultaneously. This is a great simplification over 2x665 fibers (for both S and AS regions) and single photon detectors which would need to be used instead of just a single camera sensor. At the end of each fiber an avalanche photodiode would be required and data acquisition from such a system would be a tremendously difficult.

To further underpin our argumentation here we cite the discussion from a paper by *Edgar et al.* [Nat Commun. **3**, 984 (2012)]: *(..) most measurements to date have relied on the scanning of a single avalanche photodiode or a very small number of individual detectors. This sequential scanning or use of a small number of detectors **negates any information capacity advantage in the use of spatial states.** In all protocols for high-dimensional quantum key distribution, quantum computation and teleportation using spatial states, it is **essential to perform a full field measurement of the photon transverse position**, which is made possible by using a two-dimensional (2D) detector array.*

Also the paper [Phys. Rev. Lett. **100**, 110504 (2008)] confirms the above argumentation: *(..) almost **all of these experiments employ one detector to scan through the momentum or position values**, so in principle the outcome of each measurement is binary: either the photon hits the detector or not. Therefore **this setup is not suitable for single-photon CV-QKD.** To realize the full potential of continuous variables without complex encoding, a **sufficiently large array of detectors** [avalanche photodiodes (APDs), **pixels of a CCD camera**, etc.] **is needed** to ensure that binning and truncating do not significantly diminish the information transfer rate*

Furthermore, the simultaneous observation of all photons at one camera frame is crucial for the fundamental task of achieving the violation of the EPR inequality without the *Bell sampling loophole**. Using a single fiber scanned over the entire transverse field of measurement leads to counting only a selected part of photons at each instant, leading to the results influenced by the loophole. Of course, with the 20% efficiency of our state-of-the-art camera with image intensifier, the *Bell detection loophole* would still be an unsolved problem but our idea of using camera instead of optical fibers paves the way towards future progress in this field.

* The advantage of using camera over the single-mode fibers is also discussed in [Phys. Rev. Lett. **113**, 160401 (2014)]: (...) *in most experiments the use of single photon detectors and coincidence counting leads to the detection of a very few parts of the selected photons, **generating a sampling loophole**, as well as in [Phys. Rev. A **86**, 010101(R) (2012)] paper: (...) This ensures **taking into account each detection event** that occurs in a relatively long exposure time compared to the laser pulse duration and, more importantly, compared to the coincidence time detection used in the experiments that use correlation data detection. Hence, even if there are losses and false-positive and -negative events, these spurious events are random, which is **fundamentally different from considering a priori that pairs are correlated** and detecting only in the temporal and spatial gate where the twin photon arrives.*

A part of introduction has been reframed to reflect the arguments presented above (starting from line 60): *Here we use a single-photon resolving camera for the first time to measure both correlations and auto-correlation unambiguously proving quantum character of light. (...) A two-dimensional detector is necessary to access high quantum information capacity of multimode states of light, which is unachievable with single-mode fibers. (...) Furthermore, simultaneous detection of the entire transverse field of view is essential in fundamental tests such as violation of Einstein-Podolsky-Rosen inequality without the Bell sampling loophole. At the end of Introduction we state: The experimental results presented here constitute the most essential step towards realization of the proposed protocol.*

Question 3.2

The primary application of this technique is to increase the photon generation rate while maintaining a low number of photons per mode. This, in turn, reduces the probability of accidentally generating multiple photons in any given mode which is nicely demonstrated in this work. However, another potential application (not demonstrated here) discussed in Figure 1, and elsewhere in the document, is to use active feed-forward to generate multi-photon states. This is an extremely interesting idea, but I was wondering if the authors could comment on the effect the limited read efficiency (~30%) would have on such a technique?

Answer 3.2

In the proposed idea of storing N heralded photons inside the quantum memory, taking the readout efficiency at the level $p=30\%$, the probability to generate exactly N photons in the readout process is simply p^N . It turns out one may take a lot of trials to generate N photons in the write-in process (for large N) because the probability to generate pair of photons belonging to the same spatial mode is sufficiently low to ensure true single photons generation in each of the memory mode, as long as we harness much more modes M than the desired number of photon N .

We believe the readout efficiency to be a technical limitation of the current setup. In particular, it has been recently demonstrated by *Cho et al.* ([Optica **3**, 100-107, (2016)] - cited as ref. [17] in the revised manuscript) that by employing the gradient echo memory (GEM) protocol in cold atomic quantum memory the efficiency can reach up to 87% at the single-photon level. Such efficiency, in principle achievable also in our system, would provide a significant enhancement in generating multi-photon states.

A comment on this topic has been added in the *Discussion* (line 399): *With these technical difficulties overcome, we expect that the enhanced multi-photon generation protocol we propose would be readily realizable. Integration of existing schemes with readout efficiency χ_R of nearly 0.9 and the probability to generate n photons equal $(\chi_R)^n$, will make our protocol highly competitive.*

Question 3.3

Following on from this, on line 357-9 the authors refer to the time-bandwidth product and its relevance for time-bin multiplexing. I do not completely understand what is proposed here. Does this

refer to the scheme outline in Figure 1 where multiple write pulses used to generate multiple Stokes photons before a single read pulse retrieves all of the anti-Stokes photons? If so, perhaps a reference to Figure 1 would help remind the reader.

Answer 3.3

The context in which we refer to time-bandwidth product contains two main points. The first one refers to the scheme presented in Figure 1 in which many write pulses are used to generate Stokes photons before a single read pulse retrieves (with 30% readout efficiency) all the stored photons, as was pointed by the Reviewer. The second one refers to the possibility of integrating the presented angular multiplexing solution with the temporal modes storage. To better describe these two ideas, we changed line 389 (revised manuscript numbering) to the following form: *Our quantum memory also exhibits an excellent time-bandwidth product of more than 500, which is an important figure of merit in terms of probability of retrieving all the photons stored in the memory (see Fig. 1), as well as prospective integration with time-bin multiplexing.*

Question 3.4

A final point on the multi-photon generation scheme. If this is to be successful, then it is very important that the write pulse does not undergo anti-Stokes scattering. Otherwise the excitation generated by one write pulse will be "read out" by subsequent write pulses rather than by the read pulse. I imagine that anti-Stokes scattering of the write is prohibited by selection rules by choosing appropriate circularly-polarized light? If so, this should be clarified. And what effect does the imperfect optical pumping have on this?

Answer 3.4

The strength of the Raman scattering process scales as the inverse of the detuning between laser frequency and frequency distance between energy levels involved in the transition. As is described in the *Methods*, the write and read laser beams are coupled to different atomic transitions (separated by 6800 MHz) while both lasers are resonant (read laser) or almost resonant (20 MHz for write laser), the strength of the reverse process (i.e. readout by the write beam) is negligible. In particular, following Eq. (29) from J. Mod. Opt. **63**, 2039-2047 (2016) [arXiv:1505.04118] we may calculate that the contribution to the photonic state of the term where a photon is read-out by the write pulse is of the order of $(20/6800)^2$.

The optical pumping in the $F=1$ manifold is not crucial to avoid the readout during the write-in process because of the negligible separation of m_F manifold compared to the splitting of different F sublevels. Thus neither the imperfect optical pumping nor the residual magnetic field has significant impact on the above process. Still it is extremely important to empty the $F=2$ manifold during optical pumping.

A related problem of the Stokes scattering occurring during the readout process has been analyzed in multiple cases, especially with warm atomic vapours where the detunings in the Raman processes need to be larger thus leading to significant influence of the unwanted processes of anti-Stokes scattering in write-in and mainly Stokes scattering in read-out and four-wave mixing. We are aware of the importance of such processes - see e.g. [Opt. Express **16**, 14444 (2009)], [Opt. Express **22**, 26076 (2014)] or [J. Mod. Opt. **63**, 2039-2047 (2016)].

We have added a comment on this issue in line 473: *Due to small detunings from respective energy levels the influence of deleterious processes of readout (write-in) with the write (read) laser is negligible.*

Question 3.5

In the Introduction, the authors provide a good background on the various methods for multiplexing single photon sources and QKD protocols. However, they have overlooked some important work from

the Calgary group and would suggest the following two papers be referenced: Sinclair et al. [PRL **113**, 053603 (2014)] and Grimau Puigibert et al. [arXiv: 1703.02068].

Answer 3.5

We thank the Reviewer for bringing our attention to these papers as indeed an important topic of spectral modes multiplexing was not referenced properly in the original manuscript. Please note the paper by Grimau Puigibert et al. was recently published in Phys. Rev. Lett. We add papers suggested by the Reviewer ([PRL **113**, 053603 (2014)] and [PRL **119**, 083601 (2017)]) as references [19] and [20] in the revised version of the manuscript.

Question 3.6

Line 135: The states h and g are not defined until later in the document leading to some potential confusion.

Answer 3.6

In the previous version of the manuscript the states h and g were defined only in the caption of Figure 2d (and also described in the Methods section, line 427). In the revised version of the manuscript we define the h and g states in line 150, where they appear for the first time in the spin-wave definition formula: (\dots) $g_j (h_j)$ corresponds to the $5^2S_{1/2}, F=1 (F=2)$ state of the j -th atom (\dots).

Question 3.7

Line 257: similarly the parameter κ is not defined until the "storage capabilities" section.

Answer 3.7

The symbol κ stands for the side length of the square-shaped ROI we used to analyze the data. It is defined in the line 202 of the revised version of the manuscript where it appears for the first time. Its definition was also recalled in the line 210.

Question 3.8

Figure 3a,b. Z-axis needs units. Coincidences per second?

Answer 3.8

As the coincidences presented in this figure correspond to actual total number of coincident counts per a pair of pixels (one in S arm and the other in AS arm), we believe that the most correct Z-axis label should be "Coincidences/px". We have added the legend with that label to Fig. 3. The *Methods* section now includes additional information about calibration: *The linear size of one pixel of the sCMOS camera corresponds to transverse wavevector size of 2.1 mm^{-1} or angle of $265 \mu\text{rad}$.* In the Fig. 3 caption we also remind that the measurement included 10^7 camera frames.

Other changes

We have corrected some typographical errors, in particular during the revision process we became aware that in the previous version of the manuscript the sign in the Eq. (1) was incorrect - there should be a plus rather than a minus between wavevectors in the exponent. Furthermore, in line 148 the formula erroneously featured a cross product sign "X" instead of vector dot product ".". Along with these we have corrected several grammatical mistakes in the revised version. References [39] and [64] have been updated as they had been published in *Optica* and *Applied Physics B*, respectively. A new reference cited in line 75 was added as Ref. [45].

REVIEWERS' COMMENTS:

Reviewer #1 (Remarks to the Author):

Authors have thoroughly addressed all of the concerns I had raised regarding the previous version of the manuscript. I believe the current manuscript is clearly written and it provides a significant advance in the field of quantum information and would be of interest to the broad community. Therefore, I recommend its publication in Nature Communications.

In terms of reaching close to unity retrieval efficiency, I agree with authors that it is not a fundamental limit. It only depends on OD and can reach close to 100% without significant contribution from the FWM noise in cold atomic media.

I do not completely agree with authors that the limit in generating multiple photons, simultaneously, is just the retrieval efficiency. The writing efficiency is inherently low and must be kept low to ensure two-photon events per channel are minimized. Even if detection/transmission efficiency is perfect during the writing stage, the writing probability needs to be $P_{wr} \ll 1$. In the case of n-photon states, the writing probability is thus $(P_{wr})^n$. This is a fundamental limit of the proposed scheme for generating multi-photon states, in my opinion. The total probability would be $(P_{wr})^n \times (P_{read})^n$, where P_{read} is the read efficiency that can reach close to unity as has been experimentally shown, but P_{write} is inherently low unless a cavity is used in that case it would be single mode.

Reviewers' comments and author's responses:

Reviewer #1:

We thank the Reviewer for recommending publication of our manuscript in Nature Communications. Please find below the answer to the comment about write probability on the production of multiple photons in the scheme we propose.

Authors have thoroughly addressed all of the concerns I had raised regarding the previous version of the manuscript. I believe the current manuscript is clearly written and it provides a significant advance in the field of quantum information and would be of interest to the broad community. Therefore, I recommend its publication in Nature Communications.

Question 1

In terms of reaching close to unity retrieval efficiency, I agree with authors that it is not a fundamental limit. It only depends on OD and can reach close to 100% without significant contribution from the FWM noise in cold atomic media. I do not completely agree with authors that the limit in generating multiple photons, simultaneously, is just the retrieval efficiency. The writing efficiency is inherently low and must be kept low to ensure two-photon events per channel are minimized. Even if detection/transmission efficiency is perfect during the writing stage, the writing probability needs to be $P_{wr} \ll 1$. In the case of n-photon states, the writing probability is thus $(P_{wr})^n$. This is a fundamental limit of the proposed scheme for generating multi-photon states, in my opinion. The total probability would be $(P_{wr})^n \times (P_{read})^n$, where P_{read} is the read efficiency that can reach close to unity as has been experimentally shown, but P_{write} is inherently low unless a cavity is used in that case it would be single mode.

Answer 1

We agree that indeed the writing probability p_{wr} must be kept low to maintain the high purity of generated single photons. However, the advantage of multiplexing is that we only need to keep low p_{wr} **per mode**, and thus with a high number of modes M we may achieve simultaneous high purity and high single-photon emission probability, as demonstrated. If we now consider multi-photon generation, the probability to generate exactly n photons with M modes is $\binom{M}{n} p_{wr}^n (1 - p_{wr})^{M-n}$. With small p_{wr} and large number of modes M to efficiently generate n photons in the memory, we will need to set the write efficiency to be $p_{wr} = n/M$, which in our case leads to a rough estimate of maximum 6 photons if a customary limit of $p_{wr} = 1\%$ is to be kept. Clearly higher number of angular modes or integration of other multiplexing schemes could significantly improve this figure-of-merit.